# Analysis of Flavonoid Metabolites in Chaenomeles Petals Using UPLC-ESI-MS/MS

**DOI:** 10.3390/molecules25173994

**Published:** 2020-09-02

**Authors:** Ting Shen, Fengting Hu, Qianrui Liu, Haiyan Wang, Houhua Li

**Affiliations:** 1College of Landscape Architecture and Art, Northwest A&F University, Yangling 712100, China; shenting919@163.com (T.S.); FengtingHu123@163.com (F.H.); lqr2018055899@nwafu.edu.cn (Q.L.); 2Shaanxi Academy of Forestry Sciences, Xi’an 710082, China; lgxwhy@sohu.com

**Keywords:** *Chaenomeles* petals, flavonoid metabolites, UPLC-ESI-MS/MS, OPLS-DA

## Abstract

*Chaenomeles* species are used for both ornamental decoration and medicinal purposes. In order to have a better understanding of the flavonoid profile of *Chaenomeles*, the petals of four *Chaenomeles* species, including *Chaenomeles japonica* (RB)*, Chaenomeles speciose* (ZP), *Chaenomeles sinensis* (GP), and *Chaenomeles cathayensis* (MY), were selected as experimental material. The total flavonoid content of GP was found to be the highest, followed by MY, ZP, and RB. In total, 179 flavonoid metabolites (including 49 flavonols, 46 flavonoids, 19 flavone C-glycosides, 17 procyanidins, 15 anthocyanins, 10 flavanols, 10 dihydroflavonoids, 6 isoflavones, 5 dihydroflavonols, and 2 chalcones) were identified by Ultra-Performance Liquid Chromatography-Electrospray Ionization-Tandem Mass Spectrometry. Screening of differential flavonoid metabolites showed that GP had higher levels of metabolites when compared with the other three *Chaenomeles* species. Annotation and enrichment analysis of flavonoid metabolites revealed that cyanidin 3,5-diglucoside and pelargonidin-3,5-diglucoside anthocyanins are likely responsible for the color differences of the four *Chaenomeles* petals. Additionally, a large number of flavonoids, flavonols, and isoflavones were enriched in the petals of GP. This study provides new insights into the development and utilization of *Chaenomeles* petals and provides a basis for future investigations into their utilization.

## 1. Introduction

The genus *Chaenomeles* belongs to the Rosaceae family [1], and contains multi-purpose species which are used for food, medicine and ornamental decoration. Dried fruits of the *Chaenomeles* plant were also used in traditional Chinese medicine for the treatment of sore throat, dermatophytosis, asthma, tuberculosis, diarrhea, common cold, mastitis, and hepatitis [2,3,4]. As a medicinal plant, *Chaenomeles* fruits are rich in antioxidants and useful for the treatment of influenza, tumors, liver disease, inflammation, Parkinson’s disease, and bacterial infections [5,6,7,8]. In recent years, more *Chaenomeles* plants have been planted by the food industry in order to produce juice, syrup, liquor, fruit vinegar, wine, marmalade, fruit tea, and other products [7,9,10]. Additionally, *Chaenomeles* species are important ornamental plants because of their strong environmental adaptability, as well as their appealing flowers and leaves [11,12].

Flavonoids, including anthocyanins, flavones, flavonols, and flavanols, are important chemical components of *Chaenomeles* fruits [4,13,14]. Flavonoids have been shown to play an important role in the anti-inflammatory [15], anti-cancer [16], anti-viral and anti-bacterial [17,18], anti-diabetic [19], and anti-oxidant [20] properties of many plants. Moreover, anthocyanins are responsible for many colors in plants, and therefore affect their suitability in ornamental uses.

Many flavonoids are present in *Chaenomeles* fruits, including myricetin, kaempferol, vitexin, apigenin, catechin, epicatechin, rutin, hyperin, procyanidin B1, and procyanidin B2 [6,10,13,14,21,22]. However, no detailed and systematic study of flavonoid metabolites in *Chaenomeles* petals has not be conducted. In order to make full use of *Chaenomeles* petals, there is a need for a systematic evaluation of all the natural compounds in their petals. In this study, petals from *Chaenomeles japonica* (RB), *Chaenomeles speciose* (ZP), *Chaenomeles sinensis* (GP), and *Chaenomeles cathayensis* (MY) were selected as experimental materials. Ultra-Performance Liquid Chromatography-Electrospray Ionization-Tandem Mass Spectrometry (UPLC-ESI-MS/MS) was employed to profile the flavonoid metabolites in the petals of *Chaenomeles*. These results serve to improve the current understanding of flavonoid metabolites among the four *Chaenomeles* species and provide a number of new avenues for future exploration.

## 2. Results

### 2.1. Morphological Differences among the Petals of the Four Chaenomeles Species 

Typical colors of four *Chaenomeles* petals are shown in Figure 1. ZP and MY had similar color, while RB and GP were more distinct. The petals of RB were red orange, ZP and MY were aurora red, and GP was light pink.

### 2.2. Determination of Total Flavonoid Content

Total flavonoid content in the petals of the four *Chaenomeles* species were measured and found that total flavonoid content of GP was significantly higher (*p* < 0.01) than others, reaching 481 ± 8 milligrams of rutin equivalents per 100 g fresh weight (mg RE/100 g FW) (Figure 2). This amount was nearly 3-fold higher than ZP and 5 fold higher than RB. Total flavonoid content of MY was the second highest at 444 ± 5 mg RE/100 g FW. RB had the lowest total flavonoid content, with 104 ± 10 mg RE/100 g FW.

### 2.3. Metabolic Profiling

The flavonoid metabolites of the petals of the four *Chaenomeles* species were investigated based on UPLC-ESI-MS/MS and databases. A total of 179 (GP, 159; ZP, 145; ZP, 138; MY, 137) flavonoid metabolites were identified in the petals of the four *Chaenomeles* species, including 49 flavonols, 46 flavonoids, 19 flavonoid carbosides, 17 procyanidins, 15 anthocyanins, 10 dihydroflavonoids, 10 flavanols, 6 isoflavones, 5 dihydroflavonols, and 2 chalcones (Appendix A). The results of all detected flavonoid metabolites are shown in a heatmap after homogenization (Appendix A), which revealed that there were significant differences in the metabolite levels of the four species. The content of flavonoid metabolites in the GP petals compared with ZP, MY, and RB varied greatly. By clustering all flavonoid metabolites, it was revealed that nearly half of the flavonoid metabolites in GP were present at higher levels than ZP, MY, and RB.

### 2.4. Principal Component Analysis (PCA) of Differential Flavonoid Metabolites from the Petals of the Four Chaenomeles Species

PCA is a chemometric tool that uses a small number of principal components to reveal the internal structure among multiple variables. In the PCA plot, the three biological replicates of QC samples grouped together (Figure 3a,b), which indicated that they had similar flavonoid metabolite profiles and the analysis was reliable. PC1 and PC2 explained 51.04% and 27.35% of the total sample variability, respectively (Figure 3a). PC3 explained 12.13% of the sample variability (Figure 3b). In the PCA score plot, GP was clearly separated from ZP, RB, MY in PC1, indicating that GP had a significantly different flavonoid profile. This result was consistent with the fact that the correlation between GP and the other three species was very weak (Figure 3c). Moreover, ZP, RB, and MY were also separated in PC1, but had more distinct separation in PC2 (Figure 3a). Additionally, the three replicates were tightly clustered together, which indicated that the experiment was repeatable and reliable.

### 2.5. OPLS-DA of Flavonoid Metabolites from the Petals of the Four Chaenomeles Species 

Orthogonal signal correction and partial least squares-discriminant analysis (OPLS-DA) is a multivariate statistical method used to screen out orthogonal metabolite variables that are irrelevant to categorical variables. As an important parameter for evaluating the models in OPLS-DA, Q^2^ (predictability) values greater than 0.9 indicate a model with high explanatory power [23]. The OPLS-DA values between each pairwise comparison of the four *Chaenomeles* species are shown in Figure 4. The difference between GP and RB (R^2^X = 0.994, R^2^Y = 1, Q^2^Y = 1), GP and MY (R^2^X = 0.993, R^2^Y = 1, Q^2^Y = 1), ZP and GP (R^2^X = 0.993, R^2^Y = 1, Q^2^Y = 1), ZP and MY (R^2^X = 0.968, R^2^Y = 1, Q^2^Y = 0.999), ZP and RB (R^2^X = 0.938, R^2^Y = 1, Q^2^Y = 0.999), MY and RB (R^2^X = 0.979, R^2^Y = 1, Q^2^Y = 1) are shown in Figure 4. The Q^2^ values of each pair exceeded 0.9, indicating that the OPLS-DA models were a good fit and could be used for further screening of differential flavonoid metabolites.

### 2.6. Analysis of Flavonoid Metabolites by Volcano Plots and Venn Diagrams

To gain more insight into the differential flavonoid metabolites between the petals of the four *Chaenomeles* species, differential flavonoid metabolites were filtered according to the fold change (≥2 or ≤0.5), the variable importance in the projection (VIP, >1) of OPLS-DA model and the filtering criteria. The filtering results are shown in Appendix A in full detail, and illustrated by volcano plots and Venn diagrams (Figure 5). Of the 179 flavonoid metabolites, 69 (38.55%) were significantly different between GP and RB (23 higher, 46 lower), 69 (38.55%) between GP and MY (12 higher, 57 lower), 69 (38.55%) between ZP and GP, (51 higher, 18 lower), 40 (22.35%) between ZP and MY (13 higher, 27 lower), 36 (20.11%) between ZP and RB (26 higher, 10 lower), 44 (24.58%) between MY and RB (28 higher, 16 lower). The same amount of differential flavonoid metabolites was detected when comparing GP to ZP, RB or MY. Moreover, most of the flavonoid metabolites of ZP, RB and MY were lower relative to GP (Figure 5a–c). This indicated that there were more flavonoid metabolites in petals of GP than in the other three *Chaenomeles* species.

In the intersection of the Venn diagram (Figure 5g–j), 47 common differential metabolites were found among comparison groups GP vs. RB, GP vs. MY, and ZP vs. GP. Moreover, 9 were found amongst comparison groups ZP vs. GP, ZP vs. RB, and ZP vs. MY, 15 amongst comparison groups ZP vs. RB, MY vs. RB, and GP vs. RB and 15 among comparison groups GP vs. MY, ZP vs. MY, and MY vs. RB. Additionally, each comparison group had unique differential metabolites, implying that differential metabolites could clearly distinguish the four *Chaenomeles* petals from each other.

### 2.7. Functional Annotation and Enrichment Analysis of Differential Flavonoid Metabolites

The differential flavonoid metabolites of each comparison group were annotated by searching against the Kyoto Encyclopedia of Genes and Genomes (KEGG) database (https://www.kegg.jp/) in order to obtain detailed pathway information (Appendix A). The KEGG pathway classification results indicated that flavonoid metabolites which were present in differential amounts among the species sampled were mainly involved in “flavone and flavonol biosynthesis”, “anthocyanin biosynthesis”, “isoflavonoid biosynthesis”, “biosynthesis of secondary metabolites”, “biosynthesis of phenylpropanoids”, “flavonoid biosynthesis”, and “metabolic pathways” (Figure 6). The differential flavonoid metabolites present in comparisons with GP were mainly involved in “flavone and flavonol biosynthesis”, implying that flavones and flavonols distinguish GP from the other three *Chaenomeles* species. “Biosynthesis of secondary metabolites” was the main enrichment pathway in comparison groups ZP vs. MY and MY vs. RB.

## 3. Discussion

*Chaenomeles* species are multifunctional plants that are widely used in food, medicinal and ornamental decoration. Ornamental use is mainly based on petal color, while edible and medicinal use of *Chaenomeles* species is limited to its fruits, which are rich in flavonoids. Due to the health benefits of flavonoid consumption, a growing number of plant flowers are used to make scented tea, such as jasmine-scented tea [24], rose tea [25], peony tea [26], and chrysanthemum tea [27]. *Chaenomeles* petals are also rich in flavonoids, and this study systematically studied their flavonoid metabolite profiles in order to provide a reference for future work.

The total flavonoid content of GP petals was the highest and enrichment analysis of the flavonoid metabolites revealed that differential metabolites were mainly involved in anthocyanin biosynthesis, flavone and flavonol biosynthesis, and isoflavonoid biosynthesis (Appendix A).

### 3.1. Differential Metabolites Involved in Anthocyanin Biosynthesis

Anthocyanins play an important role in the coloration of plant petals [28]. Most blue or purple flowers contain delphinidin-based anthocyanins, and red or magenta flowers contain pelargonidin or cyanidin-based anthocyanins (Naonobu et al. 2013). In this study, a total of 15 anthocyanins and their derivatives from 4 aglycones (cyanidin, pelargonin, peonidin, and delphinidin) were detected in the petals of the four *Chaenomeles* species. Among them, pelargonin chloride, cyanidin-*O*-pentoside, pelargonin-*O*-hexoside-*O*-pentoside, cyanidin-*O*-hexoside-*O*-pentoside, cyanidin 3-rutinoside, cyanidin chloride, peonidin 3-*O*-glucoside chloride, delphinidin chloride, cyanidin *O*-syringic acid and jaceosidin, centaureidin were the first reported in *Chaenomeles* petals. Peonidin, jaceosidin, centaureidin and cyanidin-*O*-pentoside were only detected in GP petals. Jaceosidin is mainly found in the genus *Artemisia*, and jaceosidin has been shown to have anti-inflammatory activity [29], and centaureidin has been shown to have anti-inflammatory, anti-oxidant, anti-infection, and anti-tumor activities [30,31,32,33]. Compared with GP petals, pelargonidin 3,5-diglucoside and cyanidin 3,5-diglucoside were at a lower level in the petals of other species, which may account for the light pink color of GP petals (Figure 1). Moreover, pelargonidin 3,5-diglucoside in RB petals was higher when compared with the other three *Chaenomeles* species, which may be one of the reasons that it possesses red orange petals (Figure 1). Peonidin-3-*O*-glucoside in ZP petals was also higher when compared with MY.

### 3.2. Differential Metabolites Involved in Flavone and Flavonol Biosynthesis

Flavone and flavonol have many biological activities, including antioxidant, anti-HIV, anti-cancer, and anti-inflammatory [34,35,36]. In this study, a total of 46 flavones and 49 flavonols were detected. There were a large number of differential metabolites when GP petals were compared with other samples. Moreover, baicalin, apgenin, scutellarin, and narirutin displayed were present at higher levels in GP petals than in the other samples, which was consistent with the total flavonoid content of GP petals (Figure 2). Interestingly, a series of derivatives of quercetin in GP petals was present at higher levels when compared with the other three *Chaenomeles* species. Quercetin is one of the most abundant flavonoids [37], and numerous studies have found that it has many biological activities, including neuroprotection, antioxidant, and anticancer [38,39,40].

### 3.3. Differential Metabolites in the Isoflavone Biosynthetic Pathway

Isoflavones consist of a class of compounds with a core 3-benzopyrone, and are mainly found in soybeans [41]. Isoflavones have extensive biological activities, including antioxidant [42], anti-cancer [43], and protection against osteoporosis [44], cardiovascular diseases [45], and diabetes [46]. In this study, six isoflavones were detected, of which phlorizin and afzelechin were found in all samples. Genistein, prunetin, malonyglygenistin, and ononin were present at different levels in the samples. Malonyglygenistin was more abundant in GP petals than other samples, while genistein and prunetin were mainly enriched in GP petals and ononin was only found in the petals of RB and ZP.

## 4. Materials and Methods

### 4.1. Plant Material

The four *Chaenomeles* plants were cultivated at Northwest A&F University, Yangling, Shanxi, China (108°72′ E, 34°36′ N). The petals of blooming flowers in healthy plants with similar growing environments were gathered in March 2019. All samples were frozen in liquid nitrogen and stored in a refrigerator at −80 °C before extraction.

### 4.2. Sample Preparation and Extraction

Flavonoids were extracted according to the method of Han, Li [47], with some modifications. Fresh petals of the four *Chaenomeles* species were ground into powder and extracted with methanol in an ultrasonic bath at 25 °C for 30 min, then centrifuged at 12,000× *g* at 4 °C for 15 min. This step was repeated three times to ensure efficient extraction. Supernatants were condensed to a volume of 8 mL by rotary evaporation and stored at −4 °C for determination of total flavonoid content. All experiments were performed in triplicate.

*Chaenomeles* petals were freeze-dried for 36 h by a vacuum freeze-dryer before being ground into powder. One hundred milligrams of powder was weighed and dissolved in 1.0 mL methanol extract (70% methanol solution). The sample was placed in a refrigerator at 4 °C overnight. The supernatant was centrifuged at 10,000× *g* for 10 min. CNWBOND Carbon-GCB SPE Cartridge (250 mg, 3mL; ANPEL, Shanghai, China, www.anpel.com.cn/cnw) was pre-activated with 5 mL n-hexane:acetone (1:1), then 1mL of extract was added to the activated cartridge and eluted with 10 mL of n-hexane: acetone (1:1). The collected eluent was dried with nitrogen at 40 °C, dissolved in 1 mL 70% methanol solution, and then filtrated (SCAA-104, 0.22 μm pore size; ANPEL, Shanghai, China, http://www.anpel.com.cn/) before UPLC-MS/MS analysis. 

### 4.3. Determination of Total Flavonoids Content

The total flavonoid content of four *Chaenomeles* petals were measured by a modified aluminum chloride colorimetric method [48]. Next, 0.2 mL methanol extract was mixed with 5% sodium nitrite solution (0.2 mL) in a 10 mL volumetric flask, to which 10% aluminum nitrate solution (0.2 mL) was added and mixed after 6 min. Subsequently, 2 mL of 4% NaOH solution was added and the mixture was made up to the line with methanol after 6 min. After incubation for 15 min, the absorbance was measured at 510 nm with a spectrophotometer. The data were calculated and expressed as milligrams of rutin equivalent per 100 g of fresh weight (mg RE/100 g FW).

### 4.4. Ultra Performance Liquid Chromatography (UPLC) Conditions

The flavonoid metabolites of four *Chaenomeles* petals were analyzed by an UPLC-ESI-MS/MS system, which including UPLC (ultra performance liquid chromatography, Shim-pack UFLC SHIMADZU CBM30A system, www.shimadzu.com.cn/) and MS/MS (tandem mass spectrometry, Applied Biosystems 4500 Q TRAP, www.appliedbiosystems.com.cn/).

The UPLC analysis was performed with a Waters ACQUITY UPLC HSS T3 C18 column (1.8 µm, 2.1 mm × 100 mm). The solvent system was 0.04% acetic acid in water (mobile phase A) and 0.04% acetic acid in acetonitrile (mobile phase B). Oven temperature was set to 40 °C, and a flow rate of 0.40 mL/min was used. The A:B (*v*:*v*) gradient program was as follows: 100:0 (*v*:*v*) at 0 min, 5:95 (*v*:*v*) at 11.0 min, 5:95 (*v*:*v*) at 12.0 min, 95:5 (*v*:*v*) at 12.1 min, 95:5 (*v*:*v*) at 15.0 min. The effluent was alternatively connected to an ESI-triple quadrupole-linear ion trap (Q TRAP)-MS after UPLC [23,49].

### 4.5. ESI-Q TRAP-MS/MS 

Linear ion trap (LIT) and triple quadrupole (QQQ) scans were acquired on a triple quadrupole–linear ion trap mass spectrometer (Q TRAP), API 6500 Q TRAP LC/MS/MS System, equipped with an ESI Turbo Ion-Spray interface, which was operated in both positive and negative ion mode and controlled via Analyst 1.6 (AB Sciex). The ESI source operation parameters were as follows: Ion source, turbo spray; source temperature 500 °C; ion spray voltage (IS) 5500 V; ion source gas I (GSI), gas II (GSII), and curtain gas (CUR) were set at 55, 60, and 25.0 psi, respectively. The collision gas (CAD) was high. Instrument tuning and mass calibration were performed with 10 and 100 µmol/L polypropylene glycol solutions in QQQ and LIT modes, respectively. QQQ scans were acquired as multiple reaction monitoring (MRM) experiments with collision gas (nitrogen) set to 5 psi. Declustering potential (DP) and collision energy (CE) for individual MRM transitions were done with further DP and CE optimization. A specific set of MRM transitions was monitored for each period according to the metabolites eluted within this period [50,51]. Moreover, the MS/MS diagrams of some metabolites are shown in Appendix A.

### 4.6. Qualitative and Quantitative Analysis of Metabolites

Qualitative and quantitative analyses of metabolites followed the methods of [23,52]. Based on the self-built database MWDB (Metware Biotechnology Co., Ltd., Wuhan, China) and the public database of metabolite information, such as MassBank (http://www.massbank.jp/), KNAPSAcK (http://kanaya.naist.jp/KNApSAcK/), HMDB (http://www.hmdb.ca/) [53], MoTo DB (http://www.ab.wur.nl/moto/) and METLIN (http://metlin.scripps.edu/index.php) [54], the flavonoid metabolites of the samples were qualitatively and quantitatively analyzed by mass spectrometry. The characteristic ions of each substance were filtered by the triple quadrupole, and the signal intensity of the characteristic ions were obtained in the detector. The mass spectrometry file under the sample was opened with MultiaQuant software to conduct the integration and correction of chromatographic peaks. The area of each chromatographic peak represents the relative content of the corresponding substance. Finally, all the chromatographic peak area integral data were exported and saved. In order to compare the content difference of each metabolite in different samples among all the detected metabolites, we corrected the mass spectrum peaks of each metabolite detected in different samples according to the information of retention time and peak type of metabolites, and therefore the accuracy of the qualitative and quantitative analysis was further ensured.

### 4.7. Sample Quality Control Analysis

Quality control sample (QC) is a mixture of sample extracts (mix) with a concentration of 100 mg dry weight of petals per 1 mL methanol extract and 3 replicates (mix01, mix02, mix03) to analyze the repeatability of the sample under the same treatment method. Calculating the CV (coefficient of variation) value of each metabolite in these three mix repeats can be a measure of the volatility of the instrument and the stability of substance detection. In this study, 100% of the metabolites with CV value less than 0.5, more than 96% of the metabolites with CV value less than 0.3 and more than 93% of the metabolites with CV value less than 0.2. Moreover, during instrumental analysis, one quality control sample is inserted into every 10 test and analysis samples to monitor the repeatability of the analysis process. The duplication of metabolite extraction and detection, i.e., technical repetition, can be determined by overlapping display and analysis of total ion flow (TIC) diagrams of the essential spectrum detection and analysis of QC samples with different quality control. 

TIC maps from QC mass spectrometry are showed in Appendix A. The curve of the metabolites had high overlap and the retention time and peak intensity were consistent, therefore the signal stability was good when the mass spectrometer detected the same sample at different times.

### 4.8. Statistical Analysis

Three biological replicates were performed for each experiment. One-way analysis of variance (ANOVA) was performed by SPSS 23.0 (IBM Corporation, Armonk, NY, USA), and *p* < 0.01 was used as the cutoff for significant differences. Clustering analysis, PCA and OPLS-DA were carried out using R (http://www.r-project.org/). MultiQuant software was used to integrate and correct chromatographic peaks. Software Analyst 1.6.3 was used to process mass spectrometry data and Microsoft Office Excel 2019 was used to process data and draw some of the charts.

## Figures and Tables

**Figure 1 molecules-25-03994-f001:**
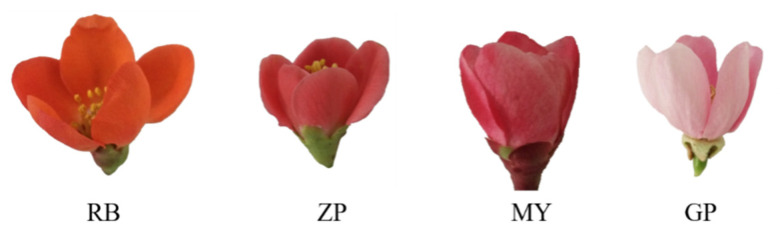
Petal colors of the four Chaenomeles plants. Chaenomeles japonica (RB), Chaenomeles speciose (ZP), Chaenomeles sinensis (GP), and Chaenomeles cathayensis (MY).

**Figure 2 molecules-25-03994-f002:**
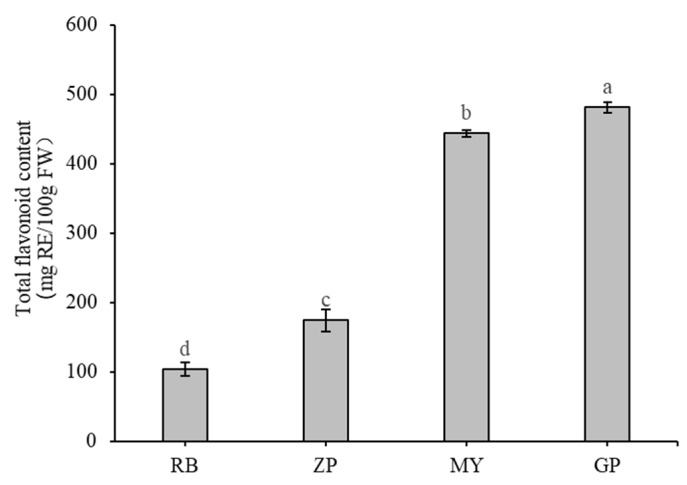
Total flavonoid content of the petals of the four *Chaenomeles* species. The lower-case letters above the histogram indicate the statistical significance at *p* < 0.01.

**Figure 3 molecules-25-03994-f003:**
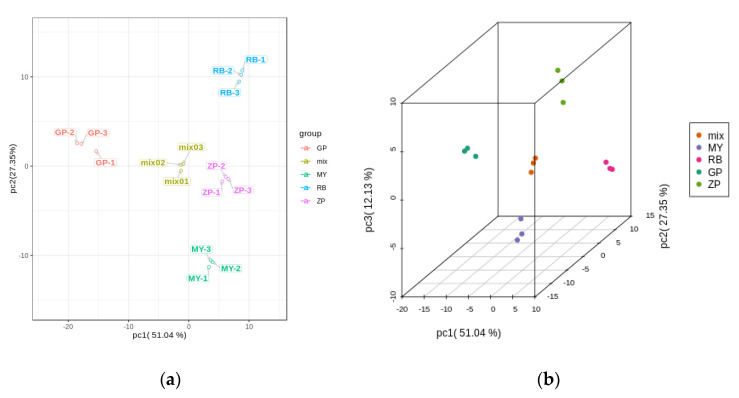
PCA score plots and correlation diagram between samples (RB, ZP, MY, GP) and quality control (QC) samples (mix). (**a**) PCA score plots; (**b**) PCA 3D score plots. Every point represents an independent biological replicate. (**c**) Correlation diagram between samples. The abscissa represents the sample name, the ordinate represents the corresponding sample name, and the color represents the value of the correlation coefficient.

**Figure 4 molecules-25-03994-f004:**
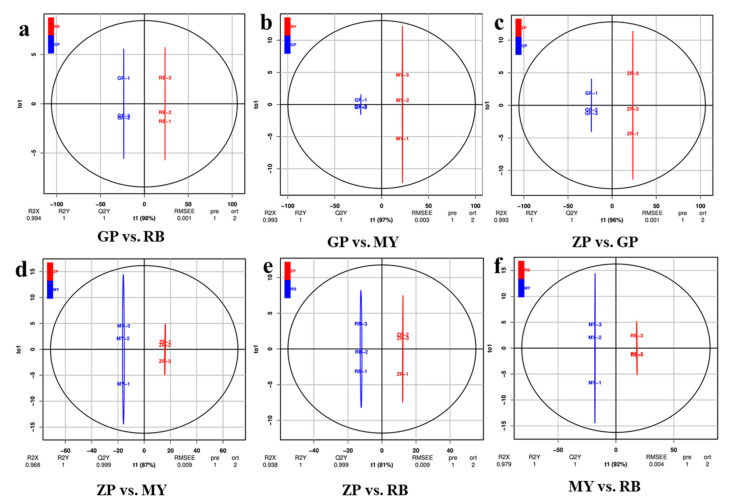
Differential flavonoid metabolite analysis on the basis of OPLS-DA. (**a**–**f**) OPLS-DA model plots for the comparison groups GP and RB, GP and MY, ZP and GP, ZP and MY, ZP and RB, MY and RB. The t1 (abscissa) represents the predicted scores of PC1, and to1 (ordinate) represents the scores of the orthogonal principal components.

**Figure 5 molecules-25-03994-f005:**
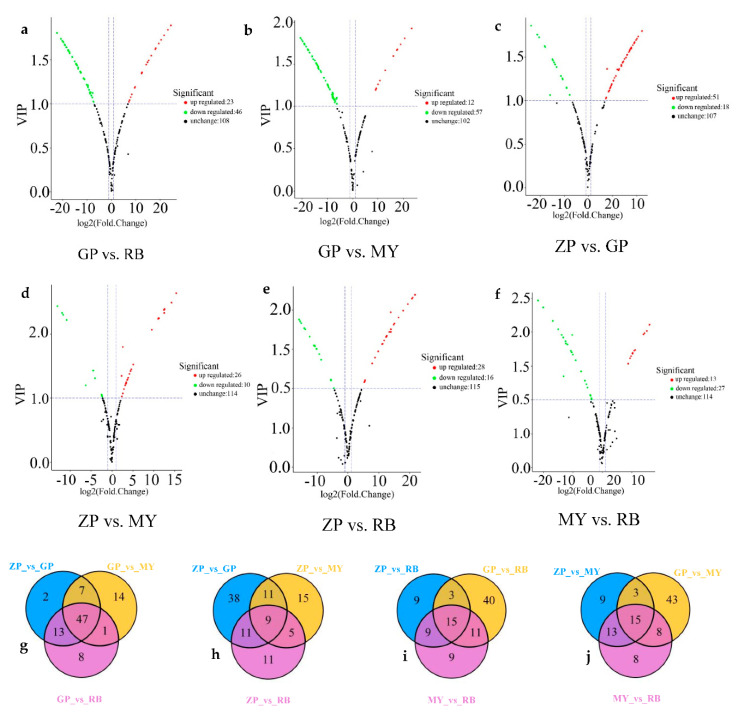
Volcano plots and Venn diagrams of flavonoid metabolites. (**a**–**f**) are volcano plots of different comparison groups: (**a**) GP vs. RB; (**b**) GP vs. MY; (**c**) ZP vs. GP; (**d**) ZP vs. MY; (**e**) ZP vs. RB; (**f**) MY vs. RB. Each point represents a metabolite. The abscissa is the logarithmic value of the fold change of each metabolite between two samples, while the ordinate is the VIP (variable important in projection) value. The greater the absolute value of the abscissa, the greater the difference in the level of the metabolite when comparing two samples. The green dots in the graph indicate metabolites with lower levels, while the red dots indicate flavonoid metabolites with higher levels. The black dots indicate the metabolites that can be detected in the sample but without significant differences among samples. (**g**–**j**) Venn diagrams of differential flavonoid metabolites in different comparison groups: (**g**) ZP vs. GP, GP vs. MY, GP vs. RB; (**h**) ZP vs. GP, ZP vs. MY, ZP vs. RB; (**i**) ZP vs. RB, GP vs. RB, MY vs. RB; (**j**) ZP vs. MY, GP vs. MY and MY vs. RB.

**Figure 6 molecules-25-03994-f006:**
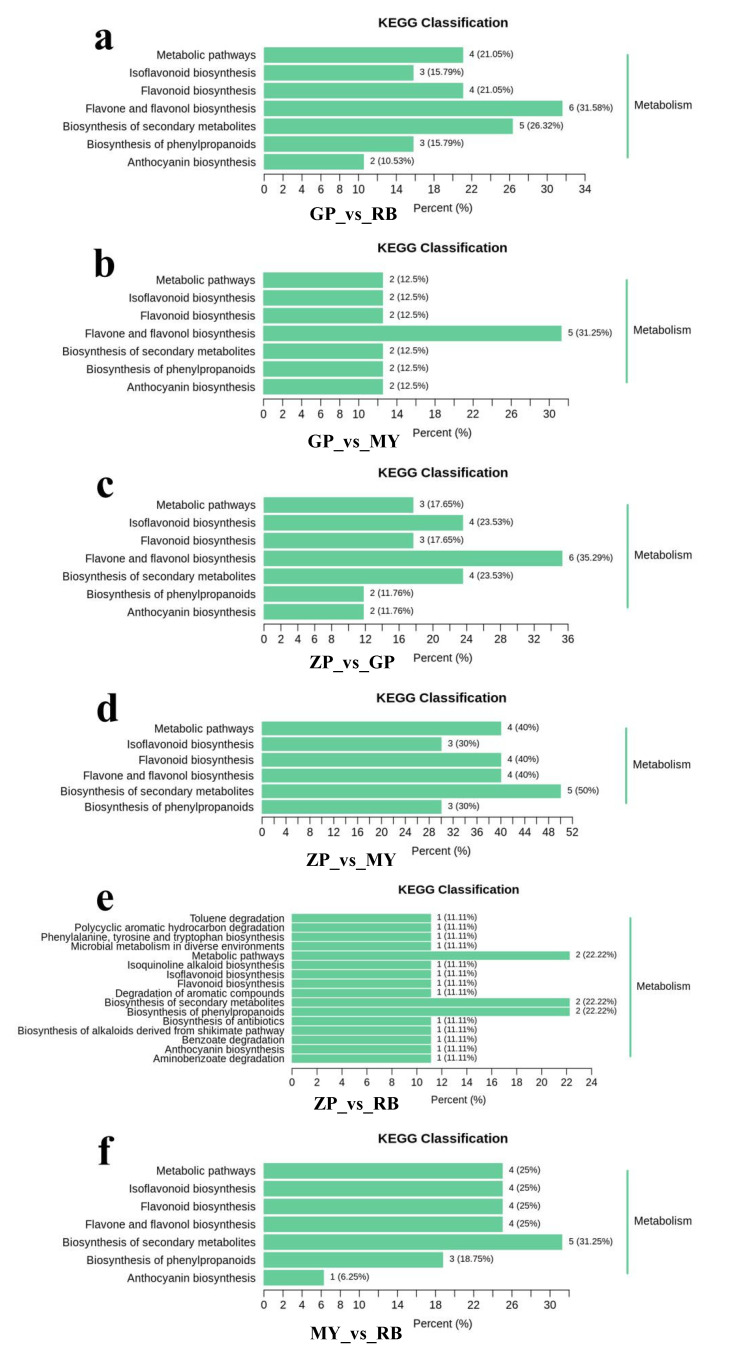
Kyoto Encyclopedia of Genes and Genomes (KEGG) pathway classification of comparison groups. (**a**) Comparison groups of GP and RB, (**b**) Comparison groups of GP and MY, (**c**) Comparison groups of ZP and GP, (**d**) Comparison groups of ZP and MY, (**e**) Comparison groups of ZP and RB, (**f**) Comparison groups of MY and RB.

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
