# Peer review of "Analysis of Flavonoid Metabolites in Chaenomeles Petals Using UPLC-ESI-MS/MS"

_molecules, 2020, doi:10.3390/molecules25173994_

Round 1

Reviewer 1 Report

This paper looks sound, and is of interest to the journal’s readership. Moderate revisions are required.  The manuscript is well structured, written and it is interesting overall.

I have some comments/questions to the authors:

- Please describe with detail all steps of the solid phase extraction procedure using CNWBOND Carbon-GCB SPE Cartridges.

- Concerning the gradient of the mobile phase, please describe what is the mobile phase to which the gradient was applied,  for example: 100:0V/V at 0min of 0.04% acetic acid in water? Or 0.04% acetic acid in acetonitrile?

- Concerning Quantitative Analysis of Metabolites please describe more details concerning how the quantification was made.  

- Information about ion suppression effects should be included in the manuscript

The coefficient of variation of the repeatability of QC should be included in the manuscript. What is the concentration of the QC?

Reviewer 2 Report

The paper is well done but the section relevant to the attributions should be deeply revised. The TIC chromatograms in figure S3 are very poor resolved and a longer gradient should be applied to separate compounds. The attributions in Table S1 are very odd. The chlorinated adduct in positive ion mode does not make any sense. I wonder how the authors made the assignments. This part should be clarified and deeply revised. Some MS/MS spectra should be inserted in the supplementary material.

The total flavonoid content and other values should be expressed with the correct number of decimal places. For instance, 443.77 ± 4.75 mg RE/100 g FW  should be 444 ± 5 mg RE/100 g FW

Author Response

Point 1: The TIC chromatograms in figure S3 are very poor resolved and a longer gradient should be applied to separate compounds.

Response 1: Thanks for your reminding. We have replaced the more high-resolution chromatograms in Figure S3. With ultra-high performance liquid column, the existing gradient can realize the rapid separation of metabolites, and the method can refer to the literature (Attachment 1).

Point 2: The attributions in Table S1 are very odd. The chlorinated adduct in positive ion mode does not make any sense. I wonder how the authors made the assignments. This part should be clarified and deeply revised.

Response 2: We are very sorry for ignoring the detailed explanation of [M-Cl]+. And we have added the following description in Table S1: “Note: the purchased compound was chloride, but there was no ion peak of chloride ion during detection. So the addition method was described as [M-Cl]+.”

Point 3: Some MS/MS spectra should be inserted in the supplementary material.

Response 3: Thank you for your reminding. We have inserted 5 MS/MS spectra in Figure S4.

Point 4: The total flavonoid content and other values should be expressed with the correct number of decimal places. For instance, 443.77 ± 4.75 mg RE/100 g FW should be 444 ± 5 mg RE/100 g FW

Response 4: We are very sorry for our negligence of number of decimal places. We have adjusted the number of decimal points in the manuscript as follows: “Total flavonoid content in the petals of the four Chaenomeles species were measured and found that total flavonoid content of GP was significantly higher (p < 0.01) than others, reaching 481 ± 8 milligrams of rutin equivalents per 100 g fresh weight (mg RE/100 g FW) (Figure 2). This amount was nearly 3-fold higher than ZP and 5 fold higher than RB. Total flavonoid content of MY was the second highest at 444 ± 5 mg RE/100 g FW. RB had the lowest total flavonoid content, with 104 ± 10 mg RE/100 g FW”.

Round 2

Reviewer 1 Report

The authors have adequalty responded to all my recommendations.

Author Response

Thank you for your reviewing.

Reviewer 2 Report

The authors reply to my concerns but still some attributions in table S1 should be revised or eliminated. Please check the assignments as [M-H]+. Do you refer to [M-H]- or [M+H]+? Please eliminate the attributions as chlorinated [M-Cl]+

Author Response

Point 1:some attributions in table S1 should be revised or eliminated. Please check the assignments as [M-H]+. Do you refer to [M-H]- or [M+H]+? Please eliminate the attributions as chlorinated [M-Cl]+.

Response 1:Thank you for your reminding, and sorry for the mistake we made here. The actual ionization mode should not have [M-H]+. We have modified [M-H]+ and eliminated the attributions as chlorinated [M-Cl]+ based on your suggestion in table S1.
